# Carbon dioxide adsorption in open nanospaces formed by overlap of saponite clay nanosheets

Kiminori Sato [1]✉ & Michael Hunger[2]

Nanoscale open spaces formed by partial overlap of two-dimensional nanosheets in clays, abundantly and ubiquitously available, possess reactive molecular sites such as nanosheet edges in their interior. Here, the capture and storage of $CO_2$ molecules in open spaces within saponite clay are explored by solid-state nuclear magnetic resonance coupled with open space analysis using positronium. $CO_2$ physisorption occurs on the nanosheet surfaces inside the open spaces under ambient conditions. Thereby, $CO_2$ molecules are activated by picking off weakly-bound oxygen from octahedral sites at the nanosheet edges and carbonate species are stabilized on the nanosheet surfaces. This instantaneous mineral carbonation and $CO_2$ physisorption occurs in the absence of an energy-consumption process or chemical solution enhancement. This finding is of potential significance for $CO_2$ capture and storage and presents an approach of environmentally friendly recycling of low contaminated soil in Fukushima.

[1] Department of Environmental Sciences, Tokyo Gakugei University, Koganei, Tokyo 184-8501, Japan. [2] Institute of Chemical Technology, University of Stuttgart, 70550 Stuttgart, Germany. ✉email: sato-k@u-gakugei.ac.jp

O wing to the significant climate change caused by the steady increase of $CO_2$ in the atmosphere along with industrial activity[1], the technology of $CO_2$ capture and storage with respect to less-energy intensiveness, cost effectiveness, and environmental friendliness has been long-awaited. In power plants, electricity is generated by burning fossil fuels such as coal and natural gas, which could be a large point source yielding ~26% of global $CO_2$ emission through the combustion process[2]. An approach to reduce $CO_2$ emission is the efficient capture of $CO_2$ and its storage in a stable manner before release to the atmosphere. There are three pathways for $CO_2$ capture: pre-combustion capture, oxyfuel combustion, and post-combustion capture[3]. Pre-combustion capture is the decarbonation process before combustion, in which primary fuels are converted into a mixture of hydrogen and $CO_2$ using gasification or reforming. In oxyfuel combustion, fuels are burned with an oxygen-enriched gas mixture instead of air, resulting in flue gases that mainly comprise $CO_2$ and $H_2O$. Post-combustion capture removes diluted $CO_2$ from the flue gases produced by the combustion of fuel in the air, which could be applied to most conventional power plants owing to its adaptable operation.

Post-combustion technology for $CO_2$ capture is based on the fundamental process of mainly chemical and physical absorption with absorbents of high $CO_2$ solubility, and physical adsorption employing solid sorbents[3]. Chemical absorption is the $CO_2$ capture process involving the reaction of $CO_2$ with chemical solvents in aqueous solution, whereas $CO_2$ is absorbed into solvents by applying pressure to promote physical absorption. In the standard environment of power plants, flue gases contain 12–14 vol.% of $CO_2$ and are emitted under atmospheric conditions, therefore, they require treatment at elevated pressure to promote $CO_2$ removal[4]. Physical adsorption utilizes the physisorption of $CO_2$ molecules onto the surface of an adsorbent via the quadrupole interaction in addition to the size effect, thus requiring microporous materials with a high specific surface area[5]. At present, physical adsorption of $CO_2$ is still in the early research stage together with the subject as to the thermal and chemical stability of the sorbed $CO_2$, mechanical strength as well as production cost.

Another important factor of $CO_2$ capture and storage is the long-term sequestration of $CO_2$, where $CO_2$ storage in geological reservoirs has been often considered[6]. In this approach, $CO_2$ is injected into a deep geological formation to be physically confined below an impermeable or very low permeability caprock, such as a shale, allowing for a sequence of possible trapping mechanisms[7]. A fraction of injected $CO_2$ is fixed as thermodynamically stable mineral carbonates, as e.g., $CaCO_3$ or $MgCO_3$ in the geological formation via the reaction with alkaline minerals there[8]. The formation of stable carbonates in the deep underground geology, known as in situ mineral carbonation[9], could be suitable for long-term storage of $CO_2$[10,11]. On the contrary, ex situ mineral carbonation is the above-ground process involving the reaction of $CO_2$ with alkaline earth metals extracted from basic rock[12]. Ex situ carbonation is generally conducted using acidic solutions at high temperature/pressure to accelerate the reaction between $CO_2$ and materials, resulting in an energy-intensive process that generates a number of liquid wastes[12,13].

Saponite, a silicate clay mineral abundantly and ubiquitously available in nature, is structured through stacks of 2D nanosheets with thicknesses of a few nm, which are the minimum structural unit. The 2D nanosheets have a variety of sizes and cannot be perfectly stacked, resulting in partial overlapping as schematically illustrated in Fig. 1a, which has been observed in images of field-emission type scanning electron microscopy[14]. This results in the formation of nanoscale open spaces (see Fig. 1b), which have been identified by positronium (Ps) annihilation spectroscopy together

with molecular dynamics simulation as is detailed later[15–17]. Naturally, there exist local molecular sites in the interior of above open spaces such as nanosheet edges that are chemically active owing to the presence of unpaired electrons at ionically bound octahedron[18]. In the present work, $CO_2$ adsorption in the open spaces originated from overlapped nanosheets in saponite clay minerals is explored by solid-state nuclear magnetic resonance (NMR) spectroscopy, open space analysis using positronium (Ps), and Fourier transform infrared (FT-IR) spectroscopy coupled with conventional chemical techniques. Besides the Na-type saponite, the Cs type is studied to find out an approach of environment-friendly recycling for low contaminated soil in Fukushima. The emergence mechanism of instantaneous mineral carbonation together with the physisorption of $CO_2$ gas molecules, feasible in the absence of energy-consumption process as well as chemical solution enhancement, is highlighted as a future strategy of $CO_2$ capture and storage.

## Results

**$CO_2$ adsorption on 2D nanosheet surfaces.** Chemical analysis by inductively coupled plasma (ICP) spectroscopy indicated that the Cs-type saponite contained ~0.04 mmol/g Cs (see $c_{cesium}$ in Table 1). According to a recently developed analytical method using the data of elution test, $^{133}Cs$ magic-angle spinning (MAS) NMR, and radiocesium interception potential, the local molecular structures, as e.g., nanosheet surface, nanosheet edge, and oncoming hexagonal cavity, have been shown to act as Cs adsorption sites[18,19]. In this analysis, 69% of the loaded Cs are found to physisorb on the surface of the 2D nanosheet, which amounts to a concentration of ~0.03 mmol/g. The carbon, hydrogen, and nitrogen (CHN) elemental analysis revealed that the C contents, $c_{carbon}$, in the Na- and Cs-type saponite samples after $CO_2$ loading are ~0.02 and ~0.18 wt.%, respectively (see Table 1). In light of the fact that the samples are isolated from the air during $CO_2$ loading, the C contents detected in the CHN analysis are solely associated with $CO_2$. Therefore, $CO_2$ molecules are sorbed to the Na- and Cs-type samples at concentrations of ~0.02 and ~0.15 mmol/g, respectively (see $c_{CO_2}$ in Table 1). The concentration of $CO_2$ higher than that of Cs on the nanosheet surface implies the adsorption of several $CO_2$ molecules at the Cs cation sites, which will be discussed more in detail later.

Ps lifetime spectroscopy prior to $CO_2$-loading reveals two kinds of open spaces for both the Na- and Cs-type saponite samples. The similar sizes of small and large open spaces with $R_1 \sim 3$ Å and $R_2 \sim 9$ Å are obtained for the Na- and Cs-type samples (see Table 2). Our former studies revealed that the above two open spaces commonly observed for both the Na- and Cs-type saponite samples are caused by overlapped nanosheets: the small and large open spaces are the consequence of one- and two-nanosheet insertion into the interlayer spaces[15]. The relative intensity of large open space $I_2$ for the Cs-type sample is ~25%, much higher than that of the Na-type sample, though the intensities $I_1$ of small open spaces at 5% are similar to each other. The high intensity $I_2$ for the Cs-type sample indicating the large amount of 9 Å open space is resultant from insufficient self-assembly toward densification, which originates from interlayer Cs cations with low hydration degree[16,17].

Figure 2 shows the $^{133}Cs$ MAS NMR spectra obtained for the Cs-type saponite before (black curve) and after $CO_2$ loading (red curve), and their difference spectrum (blue curve). The dominant signal at ~−134 ppm observed for the sample prior to $CO_2$ loading correspond to the $Cs^+$ interlayer cations physisorbed on the surfaces of the 2D nanosheets, whereas the additional broad signal at ~−15 ppm originates from $Cs_2O$ compounds at nanosheet edges[18–20]. The dominant peak arising from the $Cs^+$

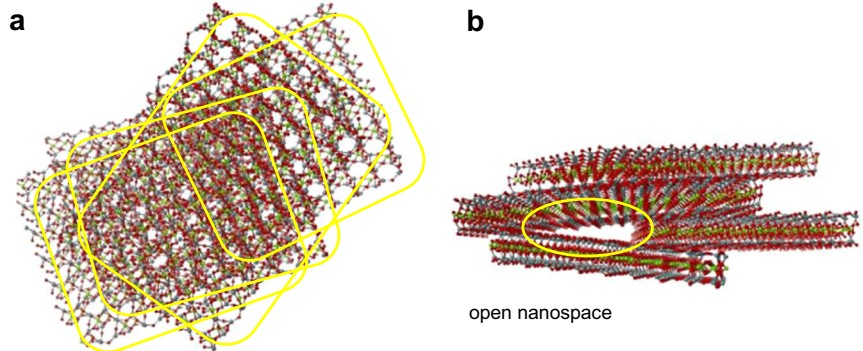

**Fig. 1 Schematic illustrations of stacked 2D clay nanosheets. a** Partial overlap viewed from the direction perpendicular to the surface of 2D nanosheets marked by solid squares. **b** Cross-section view. Note that the nanoscale open space is locally formed as marked by a solid circle. Red, gray, and green atoms correspond to oxygen, carbon, and magnesium, respectively.

**Table 1 Concentrations of loaded Cs, $c_{cesium}$, carbon, $c_{carbon}$, loaded total $CO_2$, $c_{CO2}$, $CO_2$ physisorption, $c_{Phys\_CO2}$, and $CO_2$ chemisorption, $c_{Chemi\_CO2}$.**

|  | $c_{cesium}$ (mmol/g) | $c_{carbon}$ (wt. %) | $c_{CO2}$ (mmol/g) | $c_{Phys\_CO2}$ (mmol/g) | $c_{Chemi\_CO2}$ (mmol/g) |
|---|---|---|---|---|---|
| Na type | 0.00 | 0.019 | 0.02 | 0.014 | 0.002 |
| Cs type | 0.04 | 0.183 | 0.15 | 0.134 | 0.019 |

**Table 2 Sizes of two kinds open spaces $R_1$ and $R_2$ together with their corresponding relative intensities $I_1$ and $I_2$ obtained for the Na- and Cs-type saponite samples. The error bars of R1 and R2 are ± 0.06 Å and ± 0.3 Å, respectively.**

|  | $R_1$ (Å) | $I_1$ (%) | $R_2$ (Å) | $I_2$ (%) |
|---|---|---|---|---|
| Na type | 3.2 | 6 | 9.4 | 9 |
| Cs type | 3.3 | 5 | 9.1 | 25 |

The error bars of $R_1$ and $R_2$ are ± 0.06 Å and ± 0.3 Å, respectively.

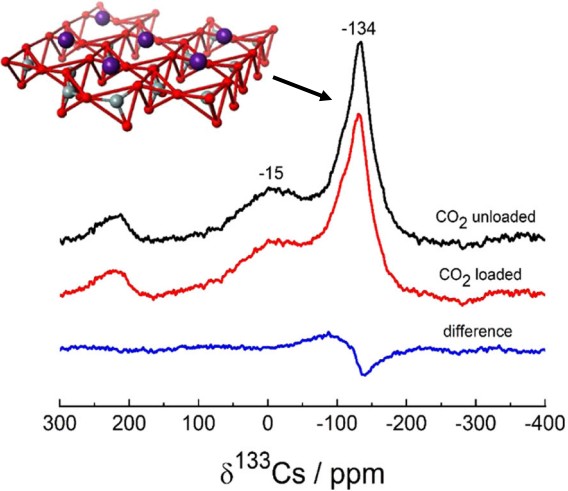

**Fig. 2 $^{133}$Cs MAS NMR spectroscopy.** $^{133}$Cs MAS NMR spectra obtained for the Cs-type saponite before (black curve) and after (red curve) $CO_2$ loading, and their difference spectrum (blue curve). The inset is schematic illustration of Cs$^+$ cations (purple) located on the surface of tetrahedral sheet consisting of oxygen (red) and carbon (gray) atoms.

interlayer cations is slightly shifted and becomes broader upon $CO_2$ loading, which can be seen in the difference spectrum as well. This demonstrates that the $CO_2$ molecules adsorb at the Cs$^+$ cation sites on the nanosheet surfaces at a concentration of ~0.03 mmol/g, as estimated above.

**Physisorption and chemisorption of $CO_2$ molecules.** Figure 3a shows the $^{13}$C MAS NMR spectra obtained for (I) Cs-type saponite $^{13}$C-enriched $CO_2$ ($^{13}CO_2$) unloaded, (II) Na-type saponite $^{13}CO_2$ loaded, (III) Cs-type saponite $^{13}CO_2$ loaded, and (IV) Cs-type saponite $^{13}CO_2$ loaded and subsequently heat treated at 200 °C for 2 h in the $N_2$ atmosphere. Before loading $^{13}CO_2$, no signal is observed in the $^{13}$C MAS NMR spectrum of the unloaded Cs-type saponite. Similarly to that, no signal appears in the NMR spectrum of unloaded Na-type saponite (data not shown here). Upon $^{13}CO_2$ loading, an intense peak and broad hump arising from $^{13}CO_2$ adsorption appeared at around the chemical shifts of 125 and 170 ppm (see (II) in Fig. 3a). This together with the above result of $^{133}$Cs MAS NMR indicates that $CO_2$ adsorption occurs at Na$^+$ cation sites on the surface of the 2D nanosheet. The dominant and additional low-field signals become intense for the Cs-type saponite (see (III) in Fig. 3a), providing the important information that the Cs-type saponite captures $CO_2$ molecules more efficiently than that in the Na-type one. After heat treatment at 200 °C for 2 h, the dominant peak disappeared, whereas the broad hump at ~170 ppm remained (see (IV) in Fig. 2a), demonstrating that the intense and small signals are ascribed to physisorption (dominant) and chemisorption (secondary) on the nanosheet surfaces, respectively. It is reasonably inferred from the study of a ternary catalyst composed of copper, zinc oxide, and alumina[21] that the broad signals at ~170 ppm originate from carbonate species, such as $Cs_2CO_3$.

It is of interest that both the physisorption and chemisorption for $CO_2$ molecules occur at the alkali metal cations located on the surface of the 2D nanosheet. It is most probable that the $CO_2$ physisorption on the nanosheet surface is caused by the quadrupole interaction between the alkali metal cations and $CO_2$ molecules. Molecular orbital calculation, on the one hand,

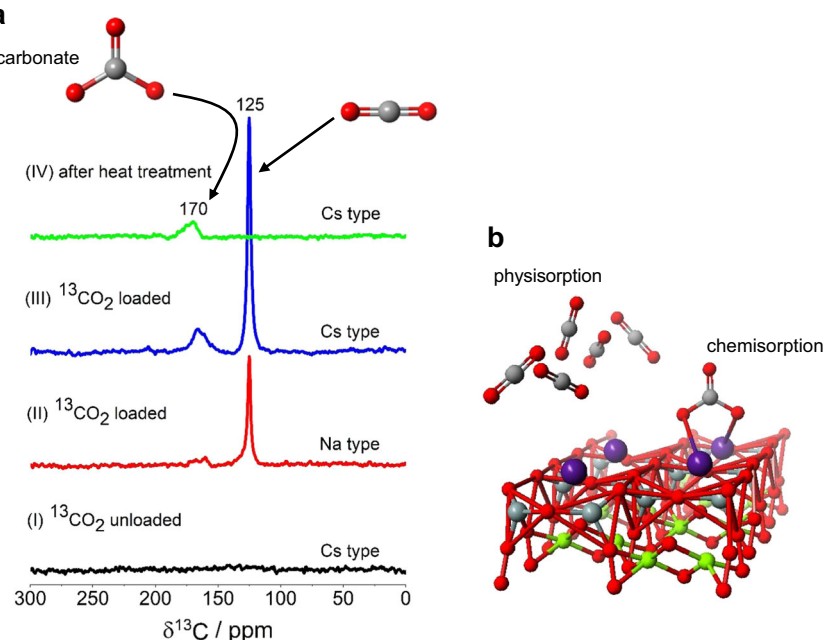

**Fig. 3 $^{13}$C MAS NMR spectroscopy. a** $^{13}$C MAS NMR spectra obtained for (I) Cs-type saponite $^{13}CO_2$ unloaded, (II) Na-type saponite $^{13}CO_2$ loaded, (III) Cs-type saponite $^{13}CO_2$ loaded, and (IV) Cs-type saponite $^{13}CO_2$ loaded and then heat treated at 200 °C for 2 h in the $N_2$ atmosphere. **b** Schematic illustration of $CO_2$ physisorption and optimized structure of $Cs_2CO_3$ as chemisorption on the nanosheet surface. Red, gray, green, and purple atoms correspond to oxygen, carbon, magnesium, and cesium, respectively.

predicts the resultant carbonate species $Cs_2CO_3$ from $CO_2$ chemisorption, in which the $CO_2$ molecule stabilizes by bridging with two alkali cations on the nanosheet surface (Fig. 3b). The adsorption energy for the optimized structure calculated with the SCIGRESS program (Fujitsu Ltd. Japan) is −97.8 kcal/mol, which is similar to that of $CO_2$ chemisorption onto the graphitic surface with two-bond conformation[22]. The concentrations of $CO_2$ physisorption, $c_{PhysCO2}$ and $CO_2$ chemisorption, $c_{ChemiCO2}$ for the Na- and Cs-type samples obtained from the total amount of $CO_2$ and the ratio of two peak intensities in $^{13}$C MAS NMR spectra are listed in Table 1. It is noted that ~13% of the loaded $CO_2$ is instantaneously chemisorbed as the carbonate species in both samples without employing strong acid solution at ambient pressure and temperature. As the concentration of $Cs^+$ cations physisorbed on the surface of the 2D nanosheet is ~0.03 mmol/g (see Table 1), the cesium carbonate formed with two $Cs^+$ cations on the nanosheet surface has the concentration of ~0.02 mmol/g. This is consistent with the concentration of $CO_2$ chemisorption, $c_{ChemiCO2}$ (see Table 1), signifying that $Cs^+$ cations on the nanosheet surface dominantly take part in the $CO_2$ chemisorption. On the other hand, the concentration of $CO_2$ physisorption, $c_{PhysCO2}$ is higher than that of $Cs^+$ cations on the nanosheet surface. Presumably, a few $CO_2$ molecules are physisorbed at $Cs^+$ cations with the distance of several angstroms (see Fig. 3b).

## Discussion

According to a number of earlier works on zeolite materials with $CO_2$ gas, the polarizing power of exchangeable alkali cations is one of the decisive factors for the capacity of $CO_2$ gas adsorption[23]. The polarizing power of cations is inversely proportional to the ionic radius of alkali metal cations. The capacity of $CO_2$ adsorption owing to the cation-quadrupole interaction thus increases as follows: $Cs^+ < Rb^+ < K^+ < Na^+ < Li^+$[23]. This is in sharp contrast with the present observation that the concentration of both the physisorption and chemisorption increase for the Cs-

type saponite. It is noted here that the fraction of open space with the size of ~9 Å for the Cs-type saponite probed by Ps is much higher than that of the Na-type one (see relative intensity $I_2$ in Table 2). The open space formed by nanosheet overlap for the Cs-type saponite offers large enough surface area to accommodate $CO_2$ molecules, thus being responsible for both the physisorption and chemisorption for $CO_2$ molecules.

Here, we discuss why the chemisorption of $CO_2$ gas molecules, i.e., ex situ mineral carbonation, instantaneously occurs in saponite samples without acid solution under the ambient condition. This unique adsorption nature is explained by the interior structure of the open nanospace characteristic for 2D materials. Saponite possess a 2:1 layered structure with 2D nanosheets consisting of tetrahedra and distorted octahedra. O and Si atoms are located at the vertices and central site of the tetrahedron, respectively. On the other hand, the O atoms or OH groups sit on the vertices of distorted octahedron, whereas a metallic Mg atom is located at the central site of octahedron (see the inset of Fig. 4). In the case of stevensite, the same family of saponite in aluminosilicate-type 2D materials, ca. 6.7% of the central Mg atoms in the octahedra are absent (see the inset of Fig. 4). An influence of Mg missing in the octahedron on the chemical bond is visible in the wavenumber region of 600–850 cm$^{-1}$ in the FT-IR spectrum of stevensite. The FT-IR spectrum for the stevensite exhibits intense and small absorption peaks at the wavenumbers of around 650 and 775 cm$^{-1}$, which can be ascribed to Si-O-Mg and $Mg_3OH$ bending vibrations, respectively[24]. The wavenumbers of the above bending vibration bands for the stevensite are lower than those of saponite, as the missing Mg atom causes lower frequencies in the both bending vibrations. It is noted here that the absorption peaks of Si-O-Mg and $Mg_3OH$ bending vibrations are red-shifted for $CO_2$ loaded saponite as well implying atom missing in the octahedron (see Fig. 4). It is reasonably inferred that the above red-shift arise from the disappearance of O atoms from the vertices of the octahedra by the analogy of stevensite as illustrated in the inset of Fig. 4.

Based on the present findings of [133]Cs and [13]C MAS NMR and FT-IR spectroscopy, we can draw the scenario of instantaneous ex situ mineral carbonation as schematically illustrated in Fig. 5. In saponite, the O atoms are weakly bound to the octahedron by ionic bonding, in contrast to the strong semi-covalent bonding of O and Si atoms in the tetrahedron. $CO_2$ gas molecules thus easily pick up the O atoms from the octahedra at the nanosheet edges in the interior of the above open spaces to be activated as the carbonate ions of $CO_3^{2-}$ type (see right-hand side of Fig. 5). The $CO_3^{2-}$ ions are then chemisorbed at the alkali metal cations on the surface of 2D nanosheets (see left-hand side of Fig. 5). The

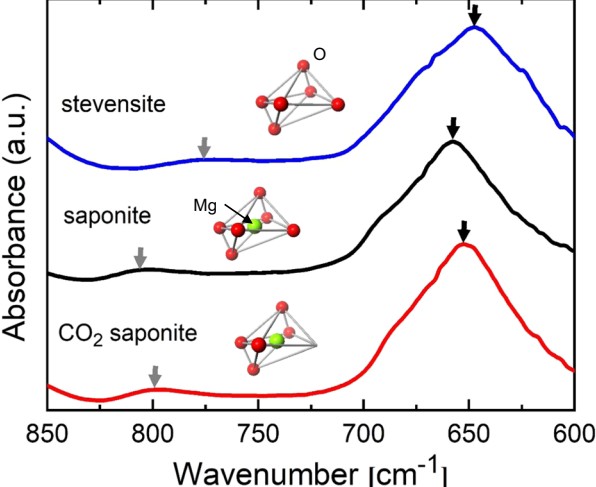

**Fig. 4 FT-IR spectroscopy.** FT-IR spectra for $CO_2$ unloaded (black curve) and loaded (red curve) saponite together with stevensite (blue curve). The absorption peaks of Si-O-Mg and $Mg_3OH$ bending vibrations are marked by black and gray thick arrows. The schematic illustrations of the disordered octahedra of stevensite, saponite, and $CO_2$-loaded saponite are shown as the insets. Note that Mg and O sites are defective.

softness of the octahedra is also anticipated from the fact that the decomposition of octahedral sheets by mechanochemical milling proceeds prior to tetrahedral sheets[25].

In the process of physical adsorption utilizing the physisorption of $CO_2$ molecules, a solid adsorbent is exposed to the combustion gas containing acid gasses such as sulfur dioxide. The central concern in the research of solid-adsorbent development is hence always a lifetime of adsorbent material relevant to thermal/chemical stability and mechanical strength in addition to the production cost. Clay minerals, such as saponite, have high tolerances for acids and their cheapness should be emphasized more than anything, thus compensating for the shortcomings of current adsorbent materials mentioned above. In light of the fact that the peak of $CO_2$ physisorption in the [13]C MAS NMR spectrum completely disappeared upon heat treatment (see Fig. 3), the open space formed by nanosheet overlap could again offer large enough surface area to accommodate $CO_2$ molecules. It is thus expected that the ability of $CO_2$ physisorption reversibly occurs when the sample is heated under mild conditions as 200 °C. In addition to that, the ability of $CO_2$ chemisorption, i.e., ex situ mineral carbonation, instantaneously emerging without chemical solution under the ambient condition is highly beneficial over amine-based systems with respect to less-energy intensiveness, cost effectiveness, and environmental friendliness. As the nuclear accident in Fukushima, the large quantities of Cs-contaminated soil containing clays generated from decontamination activities have been stored throughout prefecture. The reusing and recycling of contaminated soil on the basis of criteria for both environmental and human impacts upon proper optimization is thus awaited in the near future. The Ministry of the Environment of Japan has promoted the recycling of decontamination soil with a radioactivity level below 8000 Bq/kg for specific construction purposes as, e.g., construction of road[26]. We believe that the present findings on the ability of $CO_2$ adsorption in well-known clays could open up the recycling strategy of low contaminated soil containing clays with respect to the environment-friendly construction materials.

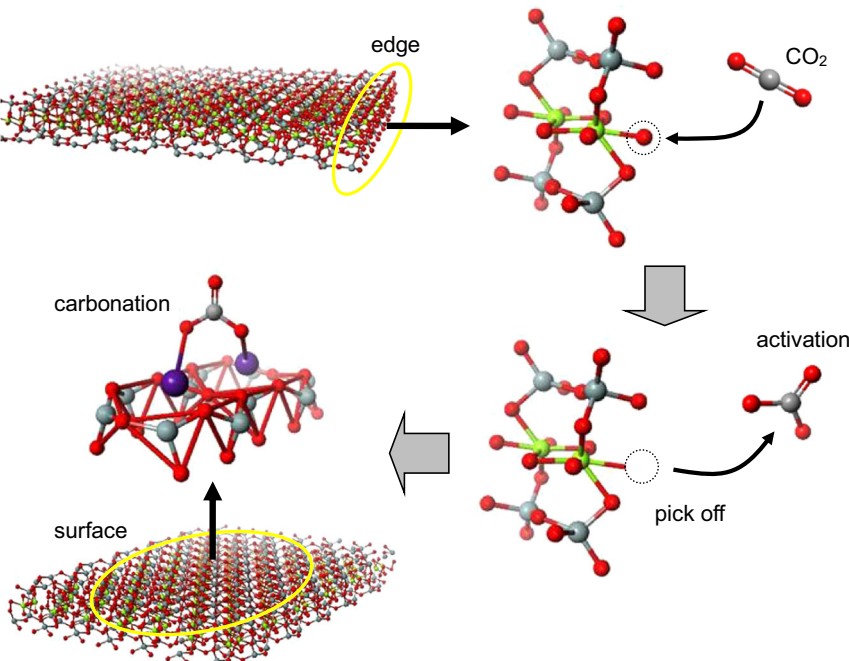

**Fig. 5 Mineral carbonation.** Scenario of instantaneous ex situ mineral carbonation under ambient conditions. Red, gray, green, and purple atoms correspond to oxygen, carbon, magnesium, and cesium, respectively.

## Methods

**Materials**. Synthetic Na-type saponite $Na_{0.66}[Mg_{5.34}Li_{0.66}]Si_8O_{20}[OH]_4$ produced by Kunimine Industries Co. Ltd., Japan, was employed in the present work. Cs loading was conducted by impregnating the Na-type saponite with a 1 M CsCl solution for ion exchanging with Cs. Aqueous solution was completely eliminated from the sample, which is referred to as Cs-type saponite. The chemical element in the Cs-type sample was examined by ICP spectroscopy (ICPS-8100, Shimadu). The sample was first treated at 423 K for 8 h under a vacuum condition of $\sim 10^{-5}$ Torr in order to eliminate physisorbed $H_2O$ molecules. The dehydrated sample was successively replaced with 1-mbar $CO_2$ atmosphere for $CO_2$ loading without any contact to air and then kept there for 30 min, which was then analyzed for CHN using an elemental analyzer and $^{133}Cs$ MAS NMR. In parallel to that, the dehydrated sample was exposed to a $^{13}C$-enriched $CO_2$ ($^{13}CO_2$) atmosphere at 50-mbar for 30 min, which was subjected to $^{13}C$ MAS NMR.

**Positronium lifetime spectroscopy**. The sizes of open nanospaces and their fractions were investigated by Ps annihilation lifetime spectroscopy employing digital oscilloscope (WaveSurfer 10, Teledyne LeCroy). The positron source ($^{22}Na$), sealed in a thin foil of Kapton, was mounted in a sample-source-sample sandwich configuration. The 1.27 MeV positron birth γ ray from a $^{22}Na$ source and one of the 511 keV γ rays emitted as a result of positron annihilation in the samples are detected by $BaF_2$ scintillators with 1" diameter × 1" thickness coupled with photomultiplier tubes (H3378-51, HAMAMATSU). Positron lifetime spectra were numerically analyzed. A fraction of energetic positrons injected into samples forms the bound state with an electron, Ps. Singlet para-Ps (p-Ps) with the spins of the positron and electron antiparallel and triplet ortho-Ps (o-Ps) with parallel spins are formed at a ratio of 1:3. Hence, three states of positrons: p-Ps, o-Ps, and free positrons exist in samples. The annihilation of p-Ps results in the emission of two γ-ray photons of 511 keV with a lifetime ~125 ps. Free positrons are trapped by negatively charged parts, such as polar elements, and annihilated into two photons with a lifetime ~450 ps. The positron in o-Ps undergoes two-photon annihilation with one of the bound electrons with a lifetime of a few ns after localization in nanospaces. The last process is known as o-Ps pick-off annihilation and provides information on the free volume size R through its lifetime $\tau_{o-Ps}$ based on the Tao-Eldrup model[27,28].

$$\tau_{o-Ps} = 0.5\left[1 - \frac{R}{R_0} + \frac{1}{2\pi}\sin\left(\frac{2\pi R}{R_0}\right)\right]^{-1} \qquad (1)$$

where $R_0 = R + \Delta R$, and $\Delta R = 0.166$ nm is the thickness of homogeneous electron layer in which the positron in o-Ps annihilates. On the one hand, the relative intensity of $\tau_{o-Ps}$ is assumed to be correlated with the amount of open nanospaces.

**Solid-state nuclear magnetic resonance**. The $^{133}Cs$ and $^{13}C$ MAS NMR experiments were performed using a Bruker Avance III 400WB spectrometer at the resonance frequencies of 52.5 MHz and 100.6 MHz, respectively. $^{133}Cs$ MAS NMR spectra were recorded with single-pulse excitation of 2.0 μs and the repetition time of 3 s. A sample spinning rate of 22 kHz was used and 24,000 scans were accumulated. The chemical shifts were referenced to a 1.0 M solution of CsCl. $^{13}C$ MAS NMR measurements were performed via exciting the $^{13}C$ spins with single-pulses of 2.0 μs and with a repetition time of 20 s thus avoiding relaxation effects by $T_1$. The sample spinning rate was 8 kHz and 320 scans were collected for each spectrum. The chemical shifts of the $^{13}C$ nuclei in the adsorbed organic species were determined with respect to tetramethylsilane as the external reference with an accuracy ±1 ppm.

**FT-IR spectroscopy**. Attenuated total reflection (ATR) FT-IR spectra were measured using a Nicolet iS5 FT-IR spectrometer (Thermo Fisher Scientific Inc.) using the ATR device with a diamond crystal plate. All the FT-IR spectra were measured at room temperature with the resolution of 4 $cm^{-1}$. The measurements were repeated 100 times and the final spectra were obtained by averaging them. OMNIC 8.2 software was used to display absorbance spectra by converting ATR data, where the absorption band in the range of wavenumbers from 600 to 850 $cm^{-1}$ is focused.

## Data availability

The data that support the findings of this study are available from the corresponding author upon reasonable request.

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

## Acknowledgements

This work was partially supported by Grants-in-Aid of the Ministry of Education, Culture, Sports, Science and Technology of Japan (Grant nos. 16K05394, 18K04884, 18KK0382, and 20K14372).

## Author contributions

K.S. and M.H. designed and conducted the present experiments. K.S. and M.H. discussed the results and contributed to writing of the manuscript.

## Competing interests

The authors declare no competing interests.
