## [Peer Review File · Communications Chemistry]

Reviewers' comments:

Reviewer #1 (Remarks to the Author):

The authors present an interesting study evaluating the ability of Cs-rich clays to serve as sorbents for CO₂ as an alternative to conventional CO₂ capture technologies. The concept addresses a need to reuse Cs-contaminated sediment from the site of the Fukushima Daiichi nuclear disaster in Japan. This is an interesting (innovative) approach for material recycling, I wonder how feasible it is from a public acceptance perspective? Regardless, the authors present a well-written and conducted study that demonstrates Cs-rich clays can sorb as much as 0.15 mmol/g CO₂, with ~15% being chemisorbed. One thing I wish they had looked at was reversibility of the physisorbed CO₂. This is a critical need to understand the stability of the sorbed CO₂ for the timeframes necessary to duplicate geologic storage. I think the manuscript would be improved if they added a discussion on this.

Overall, I found the manuscript to be well-written with sound scientific procedures conducted. I believe the manuscript adds to the scientific literature and will be of interest to the readers of Communications Chemistry. I have directly edited the manuscript (see attached file) and have included several comments that should be addressed to improve the clarity of the manuscript. Thank you for the opportunity to review this manuscript.

Reviewer #2 (Remarks to the Author):

The work entitled "Carbon dioxide adsorption in open nanospaces formed by overlap among clay nanosheets" deals with sorption of CO₂ onto two saponite clays. The results are interesting ones, but I have some special issues before it can be published on Communications Chemistry.

- The used clay must be evidenced into the title and abstract. Maybe you can change clay by saponite.

Abstract:

- include the used clay
- Regarding Fukushima contaminated soils, it contains CO₂ in higher concentrations? Why?

- Introduction title is missing

- Lines 57-58. Authors mention: "Clay, a silicate mineral abundantly and ubiquitously available in nature, is structured by a stack of 2D nanosheets with a thickness of a few nm, which is a minimum structural unit". The authors must be more specific, there are not a single clay in nature, there are many clay minerals on nature, each one with their characteristics.

- Again, in reference to the sentence "The 2D nanosheets with a variety of size cannot be perfectly stacked being partially overlapped as schematically illustrated in the left-hand side of Fig. 1, which has been observed in the images of field-emission type scanning electron microscopy (FE-SEM)¹²", said specifically for Na commercial saponite.

- Lines 64-65, it is not clear why authors mention Cs adsorption on nanosheet edges, introduction lost the conductor line.

- Line 67, please define Ps

Results

- Table 1 must be improved. It is not clear if, for example, carbon concentration is of sample before or after sorption experiments.

- Please, add the associated errors to the determined R1 and R2 values (Table 2)

- How are the relative intensity of each open space? If there are two identified open spaces, should the sum of both relative intensities not be 100%?

- Line 102, regarding the sentence "This demonstrates that the CO₂ molecules adsorb at Cs⁺ cations on the nanosheet surfaces and the concentration is reasonably assumed to be ~ 0.03 mmol/g as estimated above", what about X-Ray diffraction technique? Probably a change on the reflection peak supports this fact.

Physisorption and chemisorption of CO₂ molecules

- Why ¹³C MAS NMR spectrum of Na type saponite is not include? There are not signal as that of Cs-type clay?? Maybe a sentence could be included.

- Regarding the sentence: "Upon ¹³CO₂ loading, an intense peak and broad hump arising from CO₂ adsorption appeared at around the chemical shifts of 125 and 170 ppm (see (b) in Fig. 2). This together with the above result of ¹³³Cs MAS NMR indicates that CO₂ adsorption occurs at Na⁺ cations on the surface of 112 2D nanosheet." Figure b is the spectrum of Na-Saponite, then, why do you joint it with results of Cs-saponite? It is confusing.

Discussion

- Add some references for the sentence "According to a number of earlier works on zeolite materials with CO₂ gas, the polarizing power of exchangeable alkali cations is one of the decisive factors for the capacity of CO₂ gas adsorption"

- Regarding the sentences "the open space with the size of ~ 9 Å for the Cs-type saponite probed by Ps is much higher than that of Na-type one (see Table 2)", is confusing, do you mean that the intensity of this site is much higher for Cs-saponite than for Na-saponite?

- Please, improve figure 4 distinguish the narrows corresponding to Si-O-Mg and Mg3OH

- The reason why the red-shift of FTIR peaks of CO₂ - saponite it is not clear.

- Figure 5 must be improved indicating the color of each atom.

- Regarding the use of contaminated soils from Fukushima region, it is true that soils contain clay minerals. But, the same behavior (formation of relatively high open spaces) is expected on other clays?

Methods

- The Cs-saponite: after impregnating Na Saponite with Cs, what do you do? Is Cs replacing the Na ions? Where are now the Na ions?

- Why have you chosen Cs-saponite instead of Na-saponite for CO₂ adsorption? It is not clear. A discussion could be added into the introduction section

- How where performed the CO₂ sorption experiments?

Reply to Comments of Reviewer #1

We are grateful to you for thorough reading and constructive comments, including polishing sentences. All the criticisms made by you have been carefully considered. The manuscript was revised following your suggestions. Below, our responses to your comments. In addition, the comments suggested by Reviewer 2 have been considered. Note that revised parts in the manuscript are highlighted in red.

> One thing I wish they had looked at was reversibility of the physisorbed CO₂. This is a critical need to understand the stability of the sorbed CO₂ for the timeframes necessary to duplicate geologic storage. I think the manuscript would be improved if they added a discussion on this.

Author reply

The detailed investigation on the reversibility of CO₂ physisorption is one of our future topics, but have not yet studied. In light of the fact that the peak of CO₂ physisorption in the ¹³C MAS NMR spectrum completely disappeared upon heat treatment (see Fig. 3), the open space formed by nanosheet overlap could again offer large enough surface area to accommodate CO₂ molecules. It is thus expected that the ability of CO₂ physisorption reversibly occurs when the sample is heated under mild conditions as 200 °C. This is stated in the revised manuscript (line 12 from the bottom, page 7).

> This isn't true for basalt reservoirs. The authors need to cite the work from the CarbFix project in Iceland and BigSky project in Washington State.

Author reply

We appreciate your suggestion. This part was deleted and relevant works on CarbFix project in Iceland and Big Sky project in Washington State were alternatively cited as refs 10 and 11, respectively.

> It is unclear what you are trying to say here.

Author reply

We agree with you that this part is confusing. The phrase "characteristic for 2D structure" was deleted in the revised manuscript (see line 12, page 3).

> Concentration of what?

Author reply

We mean “site concentration”. But this part was deleted to cope with the comment from Reviewer #2.

> Can you be certain there is no contribution from organic carbon?

Author reply

A series of works relevant to CHN analysis are completely free from organic carbon. All the materials including sample holder are not associated with organic carbon. In addition, the samples were isolated from the air during CO₂ loading. We thus believe that the C contents detected in the CHN analysis are solely associated with CO₂.

> Additional details on the equipment and analysis need to be provided.

Author reply

It is now described in the revised manuscript (line 9, page 10).

> Did a physical transfer occur, or did you change the pressure in the vessel holding the sample? If a physical transfer occurred, how did you eliminate atmospheric contact?

Author reply

Here, we first evacuate the samples inside the quartz tube and then successively replace with CO₂ gas without any contact with air. This was stated in the revised manuscript (line 9, page 8)

> Again, please include information on the instrument used.

Author reply

We added the detailed information of instrument used in the present work in the revised manuscript (line 10 from the bottom, page 8).

Reply to Comments of Reviewer #2

We are grateful to you for thorough reading and constructive comments. All the criticisms made by you have been considered. The manuscript was revised following your suggestions. In addition, the comments suggested by Reviewer 1 have been carefully considered. Note that revised parts in the manuscript are highlighted in red.

> - The used clay must be evidenced into the title and abstract. Maybe you can change clay by saponite.

Author reply

The local structural unit causing CO₂ adsorption demonstrated in this work is a layered silicate, which is a main component of clay. We thus expect that any types of clay minerals can be applied for CO₂ sorbents and therefore maintained “clay” in the title.

> Abstract:

- include the used clay

Author reply

From the reason mentioned above, we maintained “clay” in the revised manuscript.

> - Regarding Fukushima contaminated soils, it contains CO₂ in higher concentrations? Why?

Author reply

Probably, our English is not good. We do not mean that contaminated soils in Fukushima contain high concentration of CO₂. We are looking for an approach of environmentally-friendly recycling for low contaminated soil in Fukushima. Thus, Cs-type saponite was studied in addition to Na-type one in the present work. This was stated in the introductory part of the revised manuscript (line 10 from the bottom, page 3).

> - Introduction title is missing

Author reply

This is our careless mistake.

“Introduction” is now placed in the revised manuscript.

> - Lines 57-58. Authors mention: “Clay, a silicate mineral abundantly and ubiquitously available in nature, is structured by a stack of 2D nanosheets with a thickness of a few nm, which is a minimum structural unit”. The authors must be more specific, there are not a single clay in nature, there are many clay minerals on nature, each one with their characteristics.

Author reply

We agree with you that a number of clay minerals are available in soil environment. Any types of clay minerals including saponite can be the target of the present CO₂ research because a layered silicate structure is of significance as mentioned above.

> - Again, in reference to the sentence “The 2D nanosheets with a variety of size cannot be perfectly stacked being partially overlapped as schematically illustrated in the left-hand side of Fig. 1, which has been observed in the images of field-emission type scanning electron microscopy (FE-SEM)¹²”, said specifically for Na commercial saponite.

Author reply

We want to emphasize again that any types of clay minerals including saponite can form the local structures partially overlapped.

> - Lines 64-65, it is not clear why authors mention Cs adsorption on nanosheet edges, introduction lost the conductor line.

Author reply

We agree with you that this part is not clear and difficult to follow. This sentence was thus deleted (see line 13, page 3).

> - Line 67, please define Ps

Author reply

This is our careless mistake.
“Ps” is now defined in the revised manuscript.

> Results

- Table 1 must be improved. It is not clear if, for example, carbon concentration is of sample before or

after sorption experiments.

Author reply

We agree with you that this part is difficult to follow. We added “after CO₂ loading” in the text of the revised manuscript (line 4, page 4).

> - Please, add the associated errors to the determined R1 and R2 values (Table 2)

Author reply

Following your suggestion, standard deviations (error bars) of R1 and R2 were added in the caption of Table 2.

> - How are the relative intensity of each open space? If there are two identified open spaces, should the sum of both relative intensities not be 100%?

Author reply

The relative intensity described in the manuscript is the fraction of Ps formation in materials. The sum of relative intensities of o-Ps, p-Ps, and free positrons can be thus 100%. Generally, the fraction of o-Ps is assumed to be correlated with the amount of open spaces. This could be confusing to general readership. An additional statement was added in the revised manuscript (line 6, page 9).

> - Line 102, regarding the sentence “This demonstrates that the CO₂ molecules adsorb at Cs⁺ cations on the nanosheet surfaces and the concentration is reasonably assumed to be ~ 0.03 mmol/g as estimated above”, what about X-Ray diffraction technique? Probably a change on the reflection peak supports this fact.

Author reply

You are right. XRD peaks are shifted toward lower diffraction angles for Cs-type saponite because of the expansion of basal spacing. We have checked XRD data, but did not employ them in the present paper due to out of the present research scope.

> Physisorption and chemisorption of CO₂ molecules

- Why ¹³C MAS NMR spectrum of Na type saponite is not include? There are not signal as that of Cs-type clay?? Maybe a sentence could be included.

Author reply

Yes, no signal appears in the NMR spectrum. This was stated in the revised manuscript (line 5, page 5).

> - Regarding the sentence: “Upon $^{13}\text{CO}_2$ loading, an intense peak and broad hump arising from CO_2 adsorption appeared at around the chemical shifts of 125 and 170 ppm (see (b) in Fig. 2). This together with the above result of ^{133}Cs MAS NMR indicates that CO_2 adsorption occurs at Na^+ cations on the surface of 112 2D nanosheet.” Figure b is the spectrum of Na-Saponite, then, why do you joint it with results of Cs-saponite? It is confusing.

Author reply

Here, we want to compared the data between Na- and Cs-types. Note that Cs-type corresponds to (c).

> Discussion

- Add some references for the sentence “According to a number of earlier works on zeolite materials with CO_2 gas, the polarizing power of exchangeable alkali cations is one of the decisive factors for the capacity of CO_2 gas adsorption”

Author reply

We should have added ref. 21 here. Ref. 21 (now 23) is now added.

> - Regarding the sentences “the open space with the size of $\sim 9 \text{ \AA}$ for the Cs-type saponite probed by Ps is much higher than that of Na-type one (see Table 2)”, is confusing, do you mean that the intensity of this site is much higher for Cs-saponite than for Na-saponite?

Author reply

Yes, we mean “intensity”.

This part could be confusing and was modified to “the fraction of open space ...” and “see relative intensity I_2 in Table 2” in the revised manuscript (line 10, page 6).

> - Please, improve figure 4 distinguish the narrows corresponding to Si-O-Mg and Mg_3OH

Author reply

Following you, the absorption peaks of Mg_3OH bending vibrations are marked by gray thick arrows. This was stated in the caption of Fig. 4 as well.

> - The reason why the red-shift of FTIR peaks of CO₂ - saponite it is not clear.

Author reply

We agree with you that this part is confusing. The paragraph was revised so that reader can see an influence of atom-missing in the octahedron on the chemical bond (see line 6 from the bottom, page 6). In addition, defective sites in the octahedron was clearly stated in the caption of Fig. 4.

> - Figure 5 must be improved indicating the color of each atom.

Author reply

Following your suggestion, the name of each atom was indicated in the caption of Fig. 5 and Fig. 1,2, and 3 as well.

> - Regarding the use of contaminated soils from Fukushima region, it is true that soils contain clay minerals. But, the same behavior (formation of relatively high open spaces) is expected on other clays?

Author reply

Yes, the same behavior is expected for other types of clay minerals as mentioned above.

> Methods

- The Cs-saponite: after impregnating Na Saponite with Cs, what do you do? Is Cs replacing the Na ions? Where are now the Na ions?

Author reply

Yes, Na cations are ion exchanged by Cs ions in CsCl aqueous solution. Na ions released in solution was eliminated from Cs-type samples by washing. This was stated in the revised manuscript (line 6, page 8).

- Why have you chosen Cs-saponite instead of Na-saponite for CO₂ adsorption? It is not clear. A discussion could be added into the introduction section

Author reply

In this work, Cs-type saponite was studied in addition to Na-type one to find out an approach of environmentally-friendly recycling for low contaminated soil in Fukushima. This is stated in the introductory part in the revised manuscript following your suggestion (line 10 from the bottom, page 3).

> - How where performed the CO2 sorption experiments?

Author reply

Here, we first evacuate the samples inside the quartz tube and then successively replace with CO2 gas without any contact with air. This was stated in in the revised manuscript (line 9, page 8)

REVIEWERS' COMMENTS:

Reviewer #1 (Remarks to the Author):

I have reviewed the revised manuscript and have found that the authors have satisfactorily addressed my comments and proposed edits. I appreciate their earnest effort to improve the readability and quality of the manuscript. I have no further edits or recommendations for them.

Reviewer #2 (Remarks to the Author):

- Regarding the previous comments about clay type must be included on title and abstract, author said: " The local structural unit causing CO₂ adsorption demonstrated in this work is a layered silicate, which is a main component of clay. We thus expect that any types of clay minerals can be applied for CO₂ sorbents and therefore maintained "clay" in the title.

It is true, the CO₂ could be then adsorbed into the interlayer space on any clay. But, it could, you are not sure, and the work deals with saponite clay. Proposing the title

"Carbon dioxide adsorption in open nanospaces formed by overlap among clay nanosheets" does not show work done, because authors only use a single type of clay for CO₂ adsorption:

Saponite. The lack of this word on title will make it difficult for interested readers to find the article. If find, the no mention in the abstract, where a short explanation about the work must be presented, does not invite to read the paper.

- Regarding the previous comments about clay type must be included on title and abstract, author said: " The local structural unit causing CO₂ adsorption demonstrated in this work is a layered silicate, which is a main component of clay. We thus expect that any types of clay minerals can be applied for CO₂ sorbents and therefore maintained "clay" in the title.

- On introduction, authors mention clays in general, without mention saponite characteristics, the used clay in this work.

- Author said that "any types of clay minerals including saponite can form the local structures partially overlapped." It cannot affirm that, works are necessary, it could be and hypothesis but not an affirmation.

- Experimental errors must be presented with significant digits, i.e., 0.22 represent an experimental error of 0.3 (Table 2).

Reply to Comments of Reviewer #2

Again, we are grateful to you for thorough reading and constructive comments. All the criticisms made by you have been carefully considered. The manuscript was revised following your suggestions. Below, our responses to your comments. Note that revised parts in the manuscript are highlighted in red.

> ... Proposing the title “Carbon dioxide adsorption in open nanospaces formed by overlap among clay nanosheets” does not show work done, because authors only use a single type of clay for CO₂ adsorption: Saponite. The lack of this word on title will make it difficult for interested readers to find the article. If find, the no mention in the abstract, where a short explanation about the work must be presented, does not invite to read the paper.

Author reply

Following you, “clay” in the title was replaced with “saponite clay”.

> ... It cannot affirm that, works are necessary, it could be and hypothesis but not an affirmation.

Author reply

Following you, “saponite” is added in the abstract. In addition, the first sentence of the second paragraph in the introductory part was revised (line 6, page 3).

> Experimental errors must be presented with significant digits, i.e., 0.22 represent an experimental error of 0.3 (Table 2).

Author reply

Following you, “0.22” in Table 2 was changed to “0.3”.